# Coupled Dynamic Analysis of a Bottom-Fixed Elastic Platform with Wave Energy Converters in Random Waves

Sanghwan Heo [1,2], Weoncheol Koo [1,]*  and Moo-Hyun Kim [2]

1  Department of Naval Architecture and Ocean Engineering, Inha University, Incheon 22212, Korea
2  Department of Ocean Engineering, Texas A&M University, Haynes Engineering Building, 727 Ross Street, College Station, TX 77843, USA
*  Correspondence: wckoo@inha.ac.kr

**Abstract:** A Wavestar-type Wave Energy Converter (WEC) on an elastic foundation structure was investigated using an author-developed coupled dynamic analysis computer program. The program included an elastic foundation structure composed of beam elements, a multi-body dynamics model of the entire system, a hydrodynamics model of the dual-buoy, and fully coupled dynamics considering the interaction between the structure and WECs. The selected WEC models a heaving-point-absorber (HPA), one of the oscillating body systems which causes rotational motions of a connecting rod attached to the foundation structure. A rotational-damper-type hydraulic power take-off (PTO) system on the foundation structure produced electricity. The bottom-fixed foundation structure was modeled by three-dimensional beam elements, and the entire system, including HPA, was analyzed by multi-body dynamics. Random wave data at Buan, a nearshore region of Korea, collected by the Korea Meteorological Administration (KMA), was used as a demonstration study using the developed computer programs. Through the case study, the displacement and stress of the foundation structure were increased significantly by the dynamic coupling effects with the WECs, which underscores that the coupled dynamic analysis is essential for a reliable performance evaluation and the design of such a system.

**Keywords:** coupled dynamic analysis; random wave; multi-body dynamics/hydrodynamics; elastic structure; wave energy converter; heaving point absorber

## 1. Introduction

Over the past few decades, numerous studies have been carried out on renewable energy to overcome the limitations/harm of fossil fuels. Among the renewable energies, ocean renewable energy is seriously considered since the ocean covers approximately 70% of the Earth's surface. In particular, ocean wave energy is abundant and has the highest energy density compared with other renewable energy sources [1,2]. Therefore, numerous studies on the potential of wave energy [1–6] and wave energy converters (WECs) converting wave energy into electrical energy [7–10] have been conducted.

WEC devices can be classified into three types according to the working principle: oscillating water columns, oscillating body systems, and overtopping systems [7]. The International Renewable Energy Agency (IRENA) reported that the installed capacity of the oscillating water column was the highest of these technologies. On the other hand, oscillating body systems are expected to play an important role in the future. In particular, the point absorber, one of the oscillating body systems, is being studied most actively because it can be scaled from small to large [11]. This study focused on a heaving-point-absorber (HPA) WEC, such as Wavestar [12]. Multiple hemispherical floating bodies were connected to the bottom-fixed main structure through hinged arms to generate electricity using the hydraulic power take-off (PTO) system.

Many studies have evaluated the dynamic response and power production of the Wavestar WEC with oscillating hemispherical bodies. Zurkinden et al. [13] performed

an experimental study on the Wavestar WEC composed of a single hemispherical float. They compared the experimental results with the BEM (boundary element method)-based numerical results. Ransley et al. [14] conducted a numerical analysis of the Wavestar WEC using a CFD-based numerical wave tank. They analyzed the dynamic response of the float and the pressure distribution on the surface of the float. Wang et al. [15] performed a numerical analysis of the point absorber WEC and arm using flexible multi-body dynamics composed of a finite element model and multi-body dynamics. They analyzed the arm stress distribution and the displacement and velocity of the float. Kim et al. [16] conducted an experimental study on a hinged hemispherical WEC. They evaluated the hydrodynamic responses and generated power through a BEM-based numerical wave tank. Windt et al. [17] performed numerical analysis and an experimental study using a CFD-based numerical wave tank for a 1/5 scale Wavestar WEC. They compared the results of the power production and the pressure on the surface of the float. Heo and Koo [18] analyzed the behavior of the two-dimensional Wavestar WECs by performing linear analysis using multi-body dynamics and classical mechanics and calculated the constraint forces acting on the hinge points. Heo and Koo [19] compared the dynamic response and generated power of Wavestar WEC considering the body-nonlinear Froude–Krylov and hydrostatic forces. The above studies focused on the behavior of the Wavestar WEC and assumed that the hinge point did not move as a center of rotation. On the other hand, the hinge point is unlikely to be fixed due to the elastic behavior of the foundation structure, particularly when the water depth was large. In turn, the movements of the floaters and the transmitted forces may affect the elastic responses and the corresponding stresses of the foundation structure significantly. These coupling effects must be analyzed to ensure the safety of the foundation structure. Studies on the combined analysis considering the movement of the hinge point connecting the WEC and the behavior of the foundation structure are very important, but not many studies have been carried out so far.

In this regard, the present study focused on the analysis of the full dynamic coupling between two floats, connecting arms with hinge points, and elastic foundation structure. For accurate analysis, a coupled analysis considering the interaction between the WEC and the structure was performed. As a demonstration site of the proposed Wavestar-system simulation, Buan, one of the nearshore regions of Korea, was selected for the detailed case study [19,20], using various sources of ocean wave data nearby [5,20–25]. The hydraulic PTO system was simplified by linear equivalent damping for the rotational motions of the arms. The foundation platform was modeled by three-dimensional beam elements, and its vibration characteristics were evaluated using modal analysis. The frequency-dependent hydrodynamic coefficients of the multiple floating bodies were obtained using a BEM-based 3D diffraction/radiation panel program, WAMIT [26]. The body-nonlinear Froude–Krylov force based on linear wave theory and body-nonlinear hydrostatic force were calculated in the time domain by considering the instantaneous submerged body surface at every time step [27,28]. Then, a fully coupled dynamic simulation was performed in the time domain to evaluate the respective body responses, relevant forces and stresses, and the corresponding power production in typical random waves of the given site. The coupled analysis results were compared with the results of several simpler approaches.

## 2. Analysis Methods

In this study, a numerical procedure was developed to perform the coupling dynamics analysis considering the interaction between the bottom-fixed elastic foundation structure and the HPA-type WECs connected with hinge points. Each WEC was assumed to be a rigid floating body, and dynamic responses were obtained using multi-body dynamics formulation considering body-nonlinear Froude–Krylov forces and hydrostatic forces. It is assumed that the bottom-fixed elastic foundation structure is composed of three-dimensional beam elements and fixed to the seabed. The interaction between the WECs and the foundation structure was considered through the constraint forces acting on the interconnected constraint points. Section 2.1 describes the numerical process of multi-body

dynamics formulation. The structural analysis process is described in Section 2.2. Irregular wave analysis to express actual sea conditions is presented in Section 2.3.

### 2.1. Equation of Motion of a Constrained Rigid Body System

#### 2.1.1. Equation of Motion

In general, a constrained rigid body system consists of motion equations of rigid bodies composed of differential equations and constraint equations composed of algebraic equations. An augmented formulation, one of the multi-body dynamics formulations, was used to calculate constraint forces acting on the constraint points and motion of each rigid floating body. The augmented formulation is a technique of expressing the equation of motion of a constrained rigid body system as differential-algebraic equations using Lagrange multipliers. The constraint equations, which are algebraic equations, were differentiated twice with respect to time and merged with the equations of motion into one matrix, as shown in Equation (1) [29].

$$\begin{bmatrix} \mathbf{M}_r & \mathbf{C}_u^T \\ \mathbf{C}_u & 0 \end{bmatrix} \left\{ \begin{array}{c} \ddot{\mathbf{u}} \\ \boldsymbol{\lambda} \end{array} \right\} = \left\{ \begin{array}{c} \mathbf{Q} \\ \mathbf{Q}_d \end{array} \right\} \tag{1}$$

where $\mathbf{u}$ is the displacement vector; $\mathbf{M}_r$ is the mass matrix (square matrix), and $\mathbf{Q}$ is the external force vector. Since each rigid body has 6 degrees of freedom, the size of the matrix and vector becomes 6 times the number of rigid bodies. The matrix on the left-hand side is a sparse matrix. If there is an interaction between different bodies, the matrix value of the corresponding degree of freedom is non-zero, and the coupling effect can be considered. $\boldsymbol{\lambda}$ is the Lagrange multiplier vector, $\mathbf{C}$ and $\mathbf{C}_u$ are constraint equations and the Jacobian matrix of constraint equations ($\mathbf{C}_u = \partial \mathbf{C}/\partial \mathbf{u}$), respectively. $\mathbf{C}_u^T$ is a transpose matrix of $\mathbf{C}_u$. $\mathbf{Q}_d$ is a vector obtained by the second derivative of the constraint equation with respect to time, as shown in Equation (2).

$$\mathbf{Q}_d = \mathbf{C}_u \ddot{\mathbf{u}} = - \left( \left( \mathbf{C}_u \dot{\mathbf{u}} \right)_u \dot{\mathbf{u}} + 2\mathbf{C}_{ut} \dot{\mathbf{u}} + \mathbf{C}_{tt} \right) \tag{2}$$

The constraint forces acting on the center of gravity of a rigid body can be calculated by multiplying the transpose of the Jacobian matrix of the constraint equations by the Lagrange multiplier vector. To obtain the constraint forces acting on the constraint points, the constraint forces acting on the center of gravity should be moved to the constraint points, and the resulting moment caused by the translational constraint forces must also be considered. Equation (3) represents the constraint forces acting on the constraint points.

$$\mathbf{Q}_{\text{joint}} = \mathbf{C}_u^T \boldsymbol{\lambda} - \left[ 0_{(3 \times 1)} \quad \left( \mathbf{u}_{jG} \times \mathbf{F}_c \right)^T \right]^T \tag{3}$$

where $\mathbf{Q}_{\text{joint}}$ is the constraint force vector acting on the joint; $\mathbf{u}_{jG}$ is the displacement vector from the constraint point to the center of gravity of the body $j$, and $\mathbf{F}_c$ is the translational constraint force vector acting on the center of gravity.

In this study, the WEC was connected to the top of a bottom-fixed structure by a massless rigid arm and was constrained by a hinge joint. Constraint equations of the hinge joint consist of three translational constraints and two rotational constraints. The translational constraints can be obtained on the condition that the position of one end of the rigid arm and the nodal point of the bottom-fixed structure are the same. For the rotational constraints, two vectors ($\mathbf{n}^{j,1}$ and $\mathbf{n}^{j,2}$) perpendicular to each other were used. These two vectors were fixed in the local coordinate system of the body $j$ and had constant values in the local coordinate system, even when the body moved, and each vector was defined to satisfy the condition that it is always perpendicular to the axis of the rotation vector. The constraint equations of the hinge joint between body $i$ and body $j$ are expressed in Equation (4) and described in Figure 1.

$$\mathbf{C}(\mathbf{u},t)_{(hinge,\,5\times1)} = \left\{ \begin{matrix} \mathbf{C}_{trans,(3\times1)} \\ \mathbf{C}_{rot,(2\times1)} \end{matrix} \right\} = \left\{ \begin{matrix} \mathbf{R}^i + \left(\mathbf{T}^i\right)^T \bar{\mathbf{u}}^i - \mathbf{R}^j - \left(\mathbf{T}^j\right)^T \bar{\mathbf{u}}^j \\ \left(\bar{\mathbf{n}}^i\right)^T \mathbf{T}^i \left(\mathbf{T}^j\right)^T \bar{\mathbf{n}}^{j,1} \\ \left(\bar{\mathbf{n}}^i\right)^T \mathbf{T}^i \left(\mathbf{T}^j\right)^T \bar{\mathbf{n}}^{j,2} \end{matrix} \right\} = 0_{(5\times1)} \quad (4)$$

where $t$ is instantaneous time; $\mathbf{R}$ is the displacement vector from the origin to the center of gravity of the body; $\mathbf{T}$ is the transformation matrix; $\bar{\mathbf{u}}$ is the displacement vector from the center of gravity to the joint; $\bar{\mathbf{n}}^i$ is the axis of the rotation vector in the local coordinates of the body $i$, $\bar{\mathbf{n}}^{j,1}$ and $\bar{\mathbf{n}}^{j,2}$ are normal vectors in the local coordinates of body $j$.

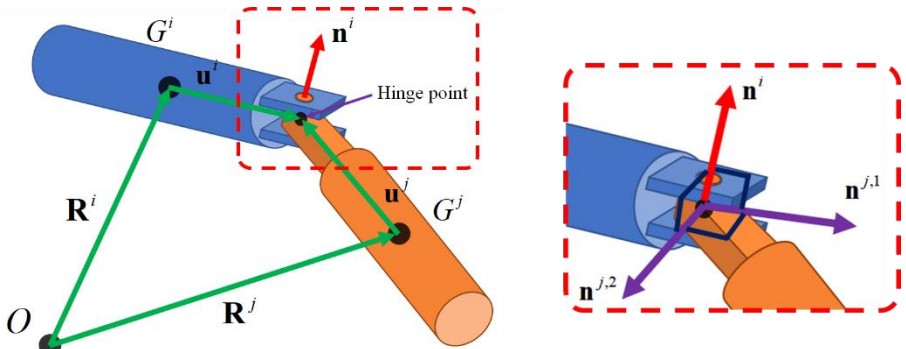

**Figure 1.** Description of a hinge joint.

The Jacobian matrix of constraint equations and $\mathbf{Q}_d$ can be derived by first and second differentiating the constraint equations with respect to time, respectively.

$$\mathbf{C}_{q\,(hinge,\,5\times12)} = \begin{bmatrix} \mathbf{I}_{3\times3} & -\left(\mathbf{T}^i\right)^T \tilde{\bar{\mathbf{u}}}^i \bar{\mathbf{G}}^i & -\mathbf{I}_{3\times3} & \left(\mathbf{T}^j\right)^T \tilde{\bar{\mathbf{u}}}^j \bar{\mathbf{G}}^j \\ 0_{2\times3} & \left(\bar{\mathbf{n}}^i\right)^T \mathbf{T}^i \left(\mathbf{T}^j\right)^T \tilde{\bar{\mathbf{n}}}^{j,n} \mathbf{T}^j \left(\mathbf{T}^i\right)^T \bar{\mathbf{G}}^i & 0_{2\times3} & -\left(\bar{\mathbf{n}}^i\right)^T \mathbf{T}^i \left(\mathbf{T}^j\right)^T \tilde{\bar{\mathbf{n}}}^{j,n} \bar{\mathbf{G}}^j \end{bmatrix} \quad (5)$$

$$\mathbf{Q}_{d\,(hinge,\,5\times1)} = \begin{bmatrix} \left(\mathbf{T}^i\right)^T \tilde{\bar{\mathbf{u}}}^i \dot{\bar{\mathbf{G}}}^i \dot{\bar{\theta}}^i - \left(\mathbf{T}^i\right)^T \left(\tilde{\bar{\omega}}^i\right)^2 \bar{\mathbf{u}}^i - \left(\mathbf{T}^j\right)^T \tilde{\bar{\mathbf{u}}}^j \dot{\bar{\mathbf{G}}}^j \dot{\bar{\theta}}^j + \left(\mathbf{T}^j\right)^T \left(\tilde{\bar{\omega}}^j\right)^2 \bar{\mathbf{u}}^j \\ -\left(\bar{\mathbf{n}}^i\right)^T \begin{pmatrix} \mathbf{T}^i \left(\mathbf{T}^j\right)^T \tilde{\bar{\mathbf{n}}}^{j,n} \mathbf{T}^j \left(\mathbf{T}^i\right)^T \dot{\bar{\mathbf{G}}}^i \dot{\bar{\theta}}^i - \mathbf{T}^i \left(\mathbf{T}^j\right)^T \tilde{\bar{\mathbf{n}}}^{j,n} \dot{\bar{\mathbf{G}}}^j \dot{\bar{\theta}}^j \\ +\left(\tilde{\bar{\omega}}^i\right)^2 \mathbf{T}^i \left(\mathbf{T}^j\right)^T \bar{\mathbf{n}}^{j,n} + \mathbf{T}^i \left(\mathbf{T}^j\right)^T \left(\tilde{\bar{\omega}}^j\right)^2 \bar{\mathbf{n}}^{j,n} - 2\tilde{\bar{\omega}}^i \mathbf{T}^i \left(\mathbf{T}^j\right)^T \tilde{\bar{\omega}}^j \bar{\mathbf{n}}^{j,n} \end{pmatrix} \end{bmatrix} \quad (6)$$

where $\tilde{\bar{\mathbf{u}}}$ is a skew-symmetric matrix of $\bar{\mathbf{u}}$, $\bar{\mathbf{G}}$ is a matrix that converts Euler angles to the angular velocity vector, $\tilde{\bar{\mathbf{n}}}$ is the skew-symmetric matrix of $\bar{\mathbf{n}}$, and $\dot{\bar{\theta}}$ is the angular velocity vector. $\tilde{\bar{\omega}}$ and $n$ are as follows:

$$\tilde{\bar{\omega}} = \mathbf{T}\left(\dot{\mathbf{T}}\right)^T, \ n = \begin{bmatrix} 1 & 2 \end{bmatrix}^T \quad (7)$$

In the time integration process, the two-loop procedure was applied to satisfy the errors of the constraint equations at the displacement, velocity, and acceleration levels within the tolerance [30–32].

### 2.1.2. External Forces Acting on a Floating Body

The external forces acting on a floating body can be expressed as Equation (8).

$$\mathbf{F}_{Float} = \mathbf{F}_{Weight} + \mathbf{F}_{Hydrostatic} + \mathbf{F}_{Froude-Krylov} + \mathbf{F}_{Diffraction} + \mathbf{F}_{Radiation} + \mathbf{F}_{Constraint} \quad (8)$$

where $\mathbf{F}_{Weight}$ and $\mathbf{F}_{Hydrostatic}$ are the self-weight and hydrostatic force on the body, respectively. $\mathbf{F}_{Froude-Krylov}$ is the Froude–Krylov force, $\mathbf{F}_{Diffraction}$ is the diffraction force, $\mathbf{F}_{Radiation}$ is the wave radiation force, and $\mathbf{F}_{Constraint}$ is the constraint force.

The self-weight acts in a vertically downward direction from the center of gravity because the float is assumed to be a rigid body. In this study, a body-nonlinear hydrostatic force considering the submerged surface of the float was calculated at every time step. The self-weight and body-nonlinear hydrostatic force of the float are presented in Equation (9).

$$\mathbf{F}_{Weight} + \mathbf{F}_{Hydrostatic} = \begin{bmatrix} 0 & 0 & -mg & 0 & 0 & 0 \end{bmatrix}^T - \iint_{S_B(t)} \rho_w g z \, \mathbf{n} \, dS \tag{9}$$

where $m$ is the mass of the float; $g$ is the gravitational acceleration; $\rho_w$ is the water density; $S_B$ is the submerged surface of the float; $z$ is the vertical displacement from the still water level; $\mathbf{n}$ is the normal vector of the body surface.

The Froude–Krylov and diffraction forces constituting the external wave force can be regarded as linear or body-nonlinear forces. The magnitude and phase angle of the linear force can be obtained through a frequency domain analysis using the boundary element method. Body-nonlinear Froude–Krylov forces must be considered to perform a precise dynamic analysis of HPA-type WECs [33]. When a body-nonlinear Froude–Krylov force is applied to a floating body, the tendency and characteristics of the dynamic response are different from when the linear Froude–Krylov force is considered [19]. The body-nonlinear Froude–Krylov force was calculated by integrating the dynamic pressure acting on the submerged surface of the body at each time step. In this study, the dynamic pressure was calculated based on the linear wave theory (or Airy wave theory).

$$\mathbf{F}_{Froude-Krylov}(t) = -\iint_{S_B(t)} \rho_w p_{FK}(t) \mathbf{n} \, dS \tag{10}$$

$$p_{FK}(t) = \frac{\rho_w g H}{2} \frac{\cosh(k(h+z))}{\cosh(kh)} \cos(k(x \cos \alpha + y \sin \alpha) - \omega t + \delta(\omega)) \tag{11}$$

where $p_{FK}$ is the dynamic pressure; $H$ is the wave height; $h$ is the water depth; $k$ is the wavenumber; $x$ and $y$ are the body surface points; $\alpha$ is the incident wave direction; $\omega$ is the wave frequency, and $\delta$ is the phase angle of the incident wave.

In the analysis of HPA-type WECs, the nonlinear wave diffraction effect can be neglected because the diameter of the floating body is very small compared to the wavelength [33]. In this study, the linear diffraction force was considered, and the magnitude and phase angle of the linear diffraction force were obtained using WAMIT, a frequency domain analysis program [26]. The time history of the linear diffraction force can be calculated through the product of the magnitude of the force obtained by frequency domain analysis and the trigonometric function as follows:

$$\mathbf{F}_{Diff}(t) = \frac{\rho_w g H}{2} \cdot \left| \mathbf{F}_{Diff}(\omega) \right| \cdot \cos(k \cdot (x \cos \alpha + y \sin \alpha) - \omega t + \varphi(\omega)) \tag{12}$$

where $\left| \mathbf{F}_{Diff} \right|$ and $\varphi$ denote the magnitude and the phase angle of linear diffraction force, respectively.

The linear wave radiation force caused by the motion of a floating body was obtained using the Cummins' equation. This equation is expressed as the convolutional integral of the velocity and retardation function of the float to account for the time memory effect [34,35].

$$F_{Radiation,l} = -m_{a_\infty,lr}(\omega) \cdot \ddot{u}_r - \int_0^t K_{lr}(\tau) \cdot \dot{u}_r(t - \tau) d\tau \tag{13}$$

where $m_{a_\infty}$ is the added mass at infinite frequency; $l$ is the direction of the radiation force; $r$ is the direction of the motion of the float. $K$ is the retardation function, as follows:

$$K_{lr}(t) = \frac{2}{\pi} \int_0^\infty b_{lr}(\omega) \cdot \cos(\omega t) d\omega \tag{14}$$

where $b$ is the radiation-damping coefficient calculated by frequency domain analysis.

### 2.2. Equation of Motion of a Bottom-Fixed Structure

In this study, the bottom-fixed structure was composed of a finite number of three-dimensional structural beam elements. The governing equation of motion of the structure is expressed as Equation (15).

$$\mathbf{M}_s\ddot{\mathbf{q}} + \mathbf{C}_s\dot{\mathbf{q}} + \mathbf{K}_s\mathbf{q} = \mathbf{F} \tag{15}$$

where $\mathbf{M}_s$, $\mathbf{C}_s$, and $\mathbf{K}_s$ are the mass, damping, and stiffness matrices of the structure, respectively. $\mathbf{q}$ is the displacement vector of the structure. $\mathbf{F}$ denotes the force vector acting on the structure, such as the hydrodynamic force, self-weight, and buoyancy.

The elements of the structure are slender bodies in which the width is much smaller than the length. The hydrodynamic force acting on the moving slender body can be obtained using the Morison equation [36].

$$d\mathbf{F} = (C_M - 1)\rho_w dV\left(\dot{\mathbf{v}} - \ddot{\mathbf{q}}\right) + \frac{1}{2}\rho_w C_D dA\left|\dot{\mathbf{v}} - \dot{\mathbf{q}}\right|\left(\dot{\mathbf{v}} - \dot{\mathbf{q}}\right) \tag{16}$$

where $dA$ and $dV$ are the infinitesimal area and volume of the body, respectively. $d\mathbf{F}$ is the wave force acting on the infinitesimal element; $C_M$ and $C_D$ are inertia coefficient and drag coefficient, respectively. $\dot{\mathbf{v}}$ is the velocity vector of water particle. The force acting on each element can be obtained by integrating Equation (8) over the entire element length.

The equation of motion presented in Equation (15) can be decoupled into $N$ independent single degree of freedom (SDOF) systems using modal analysis as follows:

$$\mathbf{I}\ddot{\mathbf{p}} + diag\left(2\beta_j\omega_j\right)\dot{\mathbf{p}} + diag\left(\omega_j^2\right)\mathbf{p} = \mathbf{\Phi}^T\mathbf{F} \tag{17}$$

where $\mathbf{I}$ is the identity matrix, $\mathbf{\Phi}$ is the modal matrix, $\mathbf{p}$ is the modal displacement vector ($\mathbf{q} = \mathbf{\Phi}\mathbf{p}$); $\beta_j$ and $\omega_j$ are the damping coefficient and natural frequency of the $j$th mode, respectively. The mode superposition method can reduce the computational time significantly, but the accuracy of the response depends on the number of modes.

In this study, two rigid floating bodies were hinged to a bottom-fixed elastic foundation structure with massless rigid arms. The constraint forces as defined in Equation (3) caused by the floating-body motions act on the hinge points of the structure. These constraint forces act as external forces at the nodal points of the structure and are included in $\mathbf{F}$ of Equation (17). The behavior of the foundation structure can affect the position, velocity, and acceleration of the hinge points, and eventually affect the motion response of the connected floating bodies. These coupling effects are discussed in Section 3.2.

### 2.3. Irregular Wave Analysis

Time history of the irregular wave was generated through the superposition of a number of linear regular waves. This study applied the two-parameter Pierson-Moskowitz (PM) spectrum to express the real sea wave condition using the following equation [37].

$$S(\omega) = \frac{173H_S^2}{T_P^4\omega^5}\exp\left(-\frac{692}{T_P^4\omega^4}\right) \tag{18}$$

where $S$ is the sea spectrum, $H_S$ is the significant wave height, and $T_P$ is the peak period. Each wave amplitude can be obtained as follows:

$$A(\omega_i) = \sqrt{2S(\omega_i)d\omega(\omega_i)} \tag{19}$$

where $d\omega$ is the frequency interval.

Under the irregular wave conditions, the wave elevation at an arbitrary location can be obtained by superposing the regular waves as follows:

$$\eta(t) = \sum_{i=1}^{Nw} A_i \cos(k_i(x\cos\alpha_i + y\sin\alpha_i) - \omega_i t + \varepsilon_i) \tag{20}$$

where $\eta$ is the wave elevation, $Nw$ is the total number of regular waves, and $\varepsilon$ is the random phase angle. The frequency interval between regular waves was randomly set to prevent the repeatability of the time history of wave elevation [38].

The time ramp function was applied to gradually increase the wave elevation and external force acting on the structure to avoid the rapid response of the numerical simulation and the instability of calculation. The ramp function is expressed as Equation (21), and $T_{ramp}$ of 50 s was applied.

$$ramp(t) = \begin{cases} 1 & for \ t \ > \ T_{ramp} \\ 1 - \cos\left(\pi \frac{t}{2T_{ramp}}\right) & for \ t \ \leq \ T_{ramp} \end{cases} \tag{21}$$

## 3. Numerical Simulations

### 3.1. Verification of Numerical Results

In the author's previous study, a fixed offshore structure was analyzed using the developed program to verify natural periods, dynamic responses, and bending stresses with commercial programs [39]. The dynamic response of the floating body was also verified through regular wave analysis and free-decay tests in the previous study [19].

In general, the motion characteristics of a floating body can be evaluated by the Response Amplitude Operator (RAO), which expresses the relationship between the wave amplitude and the body motion according to the period of the incident wave.

In this section, the RAO of the floating body for linear and body-nonlinear analysis was obtained through the time-domain simulations under irregular wave conditions, and the validity of the developed program was verified by comparing the RAO with previous studies.

The RAO was calculated using cross- and auto-spectral densities [40]. The frequency interval between regular waves was set randomly to prevent repeatability of the generated wave.

$$RAO = \frac{S_{xy}(\omega)}{S_{xx}(\omega)} \tag{22}$$

where $S_{xy}(\omega)$ and $S_{xx}(\omega)$ are the cross- and auto-spectral densities, respectively. In this process, the input signal $x(t)$ represents a time history of a random wave, and the output signal $y(t)$ represents system responses, such as the heave or pitch motion of the float. The wave was generated using the white noise spectrum shown in Figure 2 to calculate the RAO in the assigned frequency range.

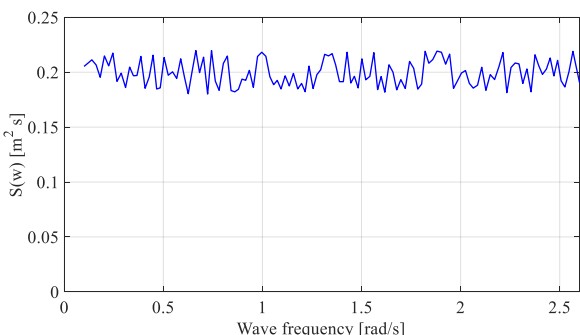

**Figure 2.** White noise spectrum to calculate the RAO of a floating body.

### 3.1.1. Heave RAO of a Spherical Float

Figure 3 depicts a spherical float with a diameter of 10 m and a draft of 5 m to verify the RAO calculation. The mass of the float was 261,800 kg. The water depth was infinite, and the density of the water was 1000 kg/m³. The float was modeled with 7200 quadrilateral elements. The float has one degree of freedom, with the motion constrained except for vertical (heave) motion. Frequency-domain hydrodynamic parameters (added mass, radiation-damping coefficient, and diffraction force) were obtained using

WAMIT. Among the external forces acting on the float, the diffraction and radiation forces were applied as linear forces.

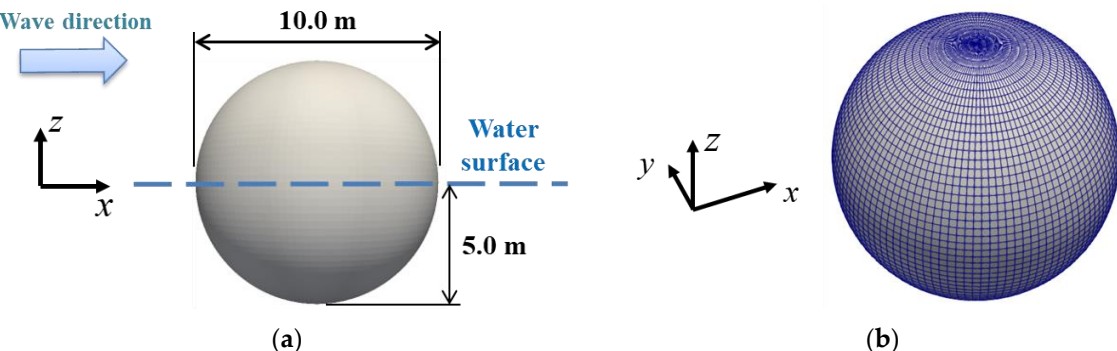

**Figure 3.** Model description of a spherical float: (**a**) Numerical model; (**b**) Panel distribution.

A time history of wave elevation was generated using the white noise spectrum shown in Figure 2. The frequency ranged from 0.1 rad/s to 4.0 rad/s, and irregular waves were generated with 196 regular component waves. A simulation result of 3600 s after a 100 s ramping period was used to remove the initial transient motions.

Figure 4 compares the heave RAOs of the half-submerged sphere with the results of a previous study [41]. While [41] obtained RAOs in regular waves of respective frequencies, the present method calculated the RAOs at all frequencies from one time-domain simulation using cross- and auto-spectral densities. The two independent results agreed well across all frequency ranges. In general, the results of the body-nonlinear analysis were similar to those of linear analysis over the entire frequency range for the given example.

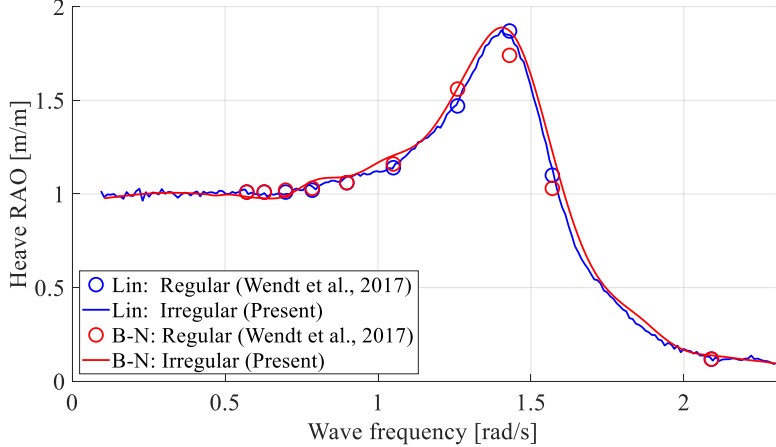

**Figure 4.** Comparison of heave RAOs of a spherical float (Lin: Linear analysis; B-N: Body-nonlinear analysis) [41].

### 3.1.2. Pitch RAO of a Spherical Float Constrained to a Hinge Point

Figure 5 depicts a spherical float constrained to a hinge point with a massless rigid arm. The diameter and draft of the float were 6 m and 5 m, respectively. The mass of the float was 57,962 kg. The water depth was 20 m, and the density of the water was 1025 kg/m$^3$. The float was modeled with 7200 quadrilateral elements. The float was constrained in motion except for the rotational motion in the $y$-axis by a hinge joint. A linear rotational damper was applied to the hinge point to describe the PTO energy extraction (damping) system. The moment acting on the hinge point ($M_{PTO}$) can be expressed as

$$M_{PTO}(t) = B_{PTO} \cdot \dot{\theta}(t) \tag{23}$$

where $\dot{\theta}$ is the angular velocity of the float, and $B_{PTO}$ is the rotational PTO damping coefficient of a rotational damper. $B_{PTO} = 3.52 \times 10^6$ N·m/(rad/s) was applied for comparison. Frequency-domain hydrodynamic parameters were also obtained using WAMIT. The diffraction and radiation forces were applied as linear forces among the external forces.

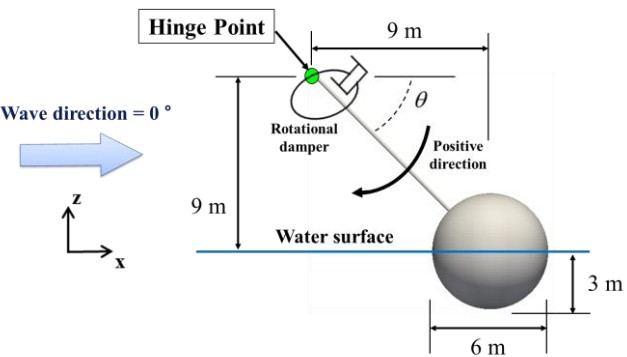

**Figure 5.** Verification model description of a spherical float constrained to a hinge point.

For this example, a time history of the incident random wave was generated using the white noise spectrum of frequency range from 0.1 rad/s to 2.6 rad/s with 126 regular component waves. A simulation result of 3600 s after a 100 s ramping period was used for the subsequent analysis.

The linear-analysis results are first presented in Figure 6a to compare the pitch RAO of the float with the previous study [42], which generally shows good agreement. Figure 6b presents the pitch RAOs obtained by authors through the body-nonlinear analysis under regular and irregular wave conditions. Both results agree well except for some minor differences in the low-frequency region. When comparing Figure 6a,b, the magnitude of the pitch RAO became smaller when the body-nonlinear analysis was performed. This trend was different from the case of Figure 4 (only heave-allowed sphere) without a rotating arm. When the floating body rotates around the hinge point, the floating body moves up and down but also in the horizontal direction, i.e., the instantaneous x-position of the body also changes. This caused the increased difference between the linear and body-nonlinear results of Figure 6b compared to Figure 4. This example illustrates that the body-nonlinear analysis can estimate the performance of the hinged-buoy system more accurately than the linear analysis.

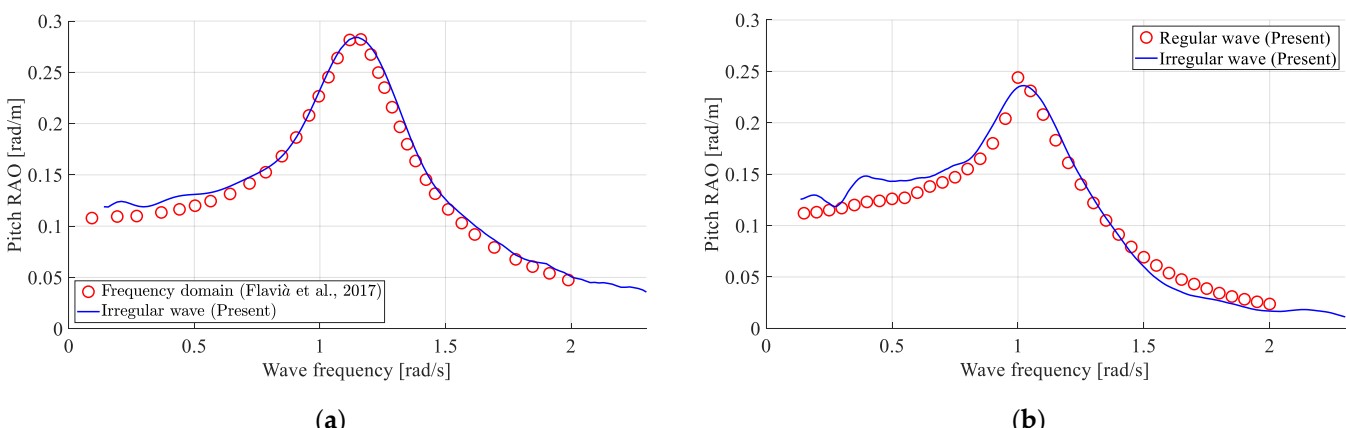

**Figure 6.** Pitch RAOs of a spherical float: (**a**) Linear analysis [42]; (**b**) Body-nonlinear analysis.

### 3.2. Simulation Results and Analysis of the Proposed System

In this study, the floating body is composed of a hemispherical shape (diameter = 2 m and draft = 1 m below the still water level) with the upper part as a vertical circular cylinder of diameter = 2 m, and height = 0.6 m. The mass of each float was 2147 kg. Each float

consists of 8520 quadrilateral elements for hydrodynamic computation. Figure 7 describes the float.

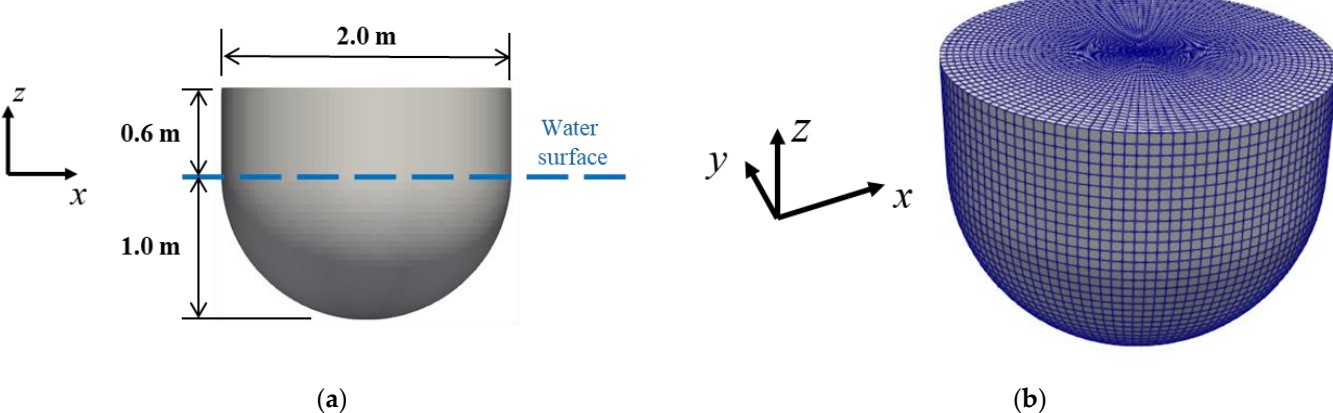

(a)  (b)

**Figure 7.** Description of a hemispherical floating body with a vertical circular cylinder on top: (**a**) Numerical model; (**b**) Panel distribution.

Before analyzing the coupled effects with the foundation structure, the pitch RAO difference between the spherical float and the present body (hemispherical float with vertical circular cylinder on top) was analyzed. The single spherical float has the same mass as the present body and consists of the same number of quadrilateral elements. The water depth was 50 m, and the applied seawater density was 1025 kg/m$^3$. Each float was fixed to a massless rigid rod, the other end of which was hinge-connected to the foundation structure. This connection allowed rotational movement with respect to the joint at the foundation structure as the center of rotation. The rotational damping proportional to the rotational velocity was applied to the hinge point to describe the PTO system, as shown in Equation (23). This PTO moment resists the rotational motion. The extraction coefficient of 100 kN·m/(rad/s) was applied through a parametric study for a regular wave [19]. Figures 8 and 9 compare their pitch RAOs, showing the body-nonlinear results. The pitch RAOs of the present body were larger than those of the sphere when the wave frequency was larger than 0.5 rad/s, i.e., in practical sea states, and the present body was superior to the sphere. In this regard, the present body shape was chosen for the subsequent section.

In addition, the pitch RAOs of the dual present bodies were calculated and compared. Figure 8 shows the set-up of single and dual floating bodies. The rotational displacement of the floating body was affected by the propagation direction of the incident wave. When the incident wave propagated in the x-direction parallel to the arm, the rotational displacement of the floating body became a maximum [19], which was assumed in this paper. The weather-side buoy was called the "front", and the lee-side buoy was called the "rear". In the case of dual buoys, the full hydrodynamic interactions of the two floating bodies were included.

Figure 9 shows the pitch RAOs of various cases. The pitch RAOs of the front and rear tend to be similar, but the rear becomes slightly larger when the wave frequency is larger than 0.7 rad/s. The differences between the single and dual were small, suggesting that the two-body interaction effects were insignificant in this example. For multiple WECs, more detailed research on the optimal arrangement needs to be investigated for maximum energy extraction efficiency [43].

In the remainder of this study, the fully coupled dynamic analysis of a bottom-fixed elastic foundation platform with WECs attached was performed. For the real sea conditions, Buan, one of the nearshore regions of Korea, was selected by referring to previous studies [19,25]. Figure 10 shows the location of the wave buoy installed in the Buan site for wave data measurement. The five-year average winter wave data were collected by the Korea Meteorological Administration (KMA) from November to February, from 2017 to 2021 [44].

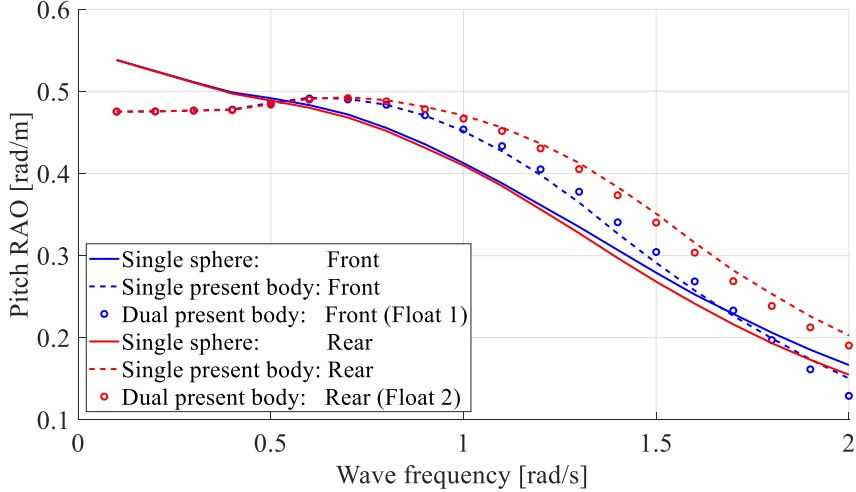

**Figure 8.** Description of single and dual floating bodies: (**a**) Single sphere: front; (**b**) Single present body: front; (**c**) Top view of single sphere/present body: front; (**d**) Single sphere: rear; (**e**) Single present body: rear; (**f**) Top view of a single sphere/present body: rear; (**g**) Side view of a dual present body; (**h**) Top view of a dual present body.

**Figure 9.** Pitch RAOs of various floating bodies by the present body-nonlinear method.

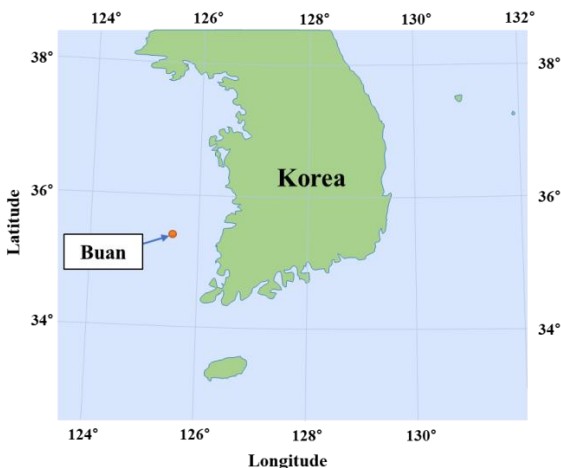

**Figure 10.** Wave buoy location in the Korean nearshore area: Buan.

Table 1 lists the average significant wave height, average peak period, and percentage of overall availability. The direction of all incident waves was the same to estimate the potential maximum power production. Based on the actual sea data, a representative significant wave height of 1.28 m and a peak period of 5.46 s (=1.15 rad/s) were selected. The water depth was 50 m, and the applied seawater density was 1025 kg/m$^3$.

**Table 1.** Five-year average winter wave data in Buan measured by the KMA [44].

| Year | Average Significant Wave Height [m] | Average Peak Period [s] | Overall Availability [%] |
|------|------|------|------|
| 2017 | 1.387 | 5.65 | 99.41 |
| 2018 | 1.211 | 5.28 | 98.75 |
| 2019 | 1.150 | 5.24 | 93.96 |
| 2020 | 1.182 | 5.33 | 89.93 |
| 2021 | 1.446 | 5.80 | 99.34 |

A PM spectrum was applied to generate a time history of the incident wave. The wave direction was fixed in the *x*-axis direction, and 130 regular component waves were superposed. The frequency range was 0.7 rad/s to 3.4 rad/s, excluding values less than 1% of the maximum spectrum value under the corresponding environmental condition. A simulation result of 3600 s after a 100-s ramping period was used for further analysis. Figure 11a shows the generated incident wave time series applied in this study. For validation, the theoretical PM spectrum and the regenerated spectrum obtained using the fast Fourier transform (FFT) with smoothing from the time series were compared (Figure 11b).

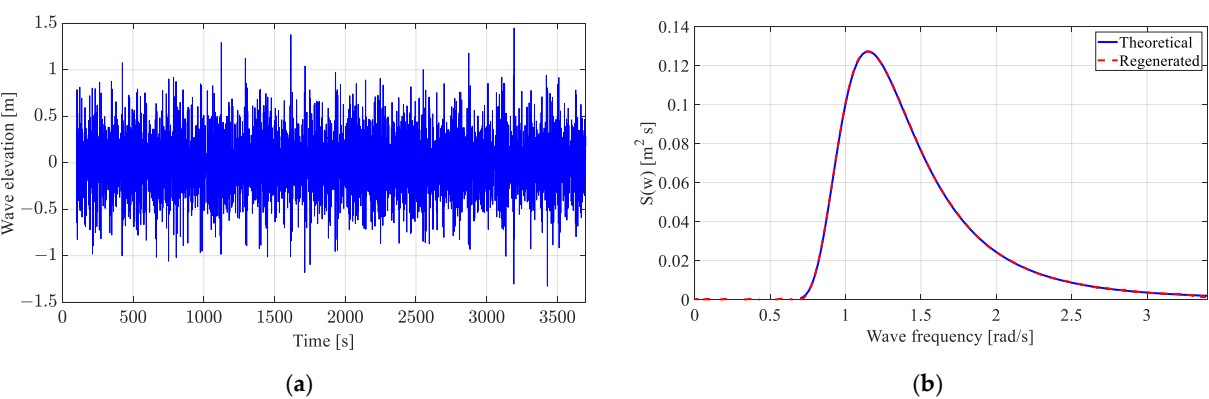

**(a)**　　　　　　　　　　　　　　　**(b)**

**Figure 11.** Description of the input wave: (**a**) Time history of wave elevation; (**b**) Wave spectrum.

Figure 12 shows the bottom-fixed foundation structure with two WECs, and Table 2 lists the detailed structural properties of the platform. The platform consisted of 122 nodal points and 124 three-dimensional beam elements, and the deck was supported by four legs. The nodal points at the bottom are fixed to the seabed. The structural elements were all slender, and the Morison equation could be used. The inertia and drag coefficients of each leg were 2.0 and 1.0, respectively, which is typical for such slender cylindrical elements. The deck weight was 980 kN, and it acted equally on each leg. Two floating buoys in Figure 7 were connected to the foundation structure, as described in the previous sections.

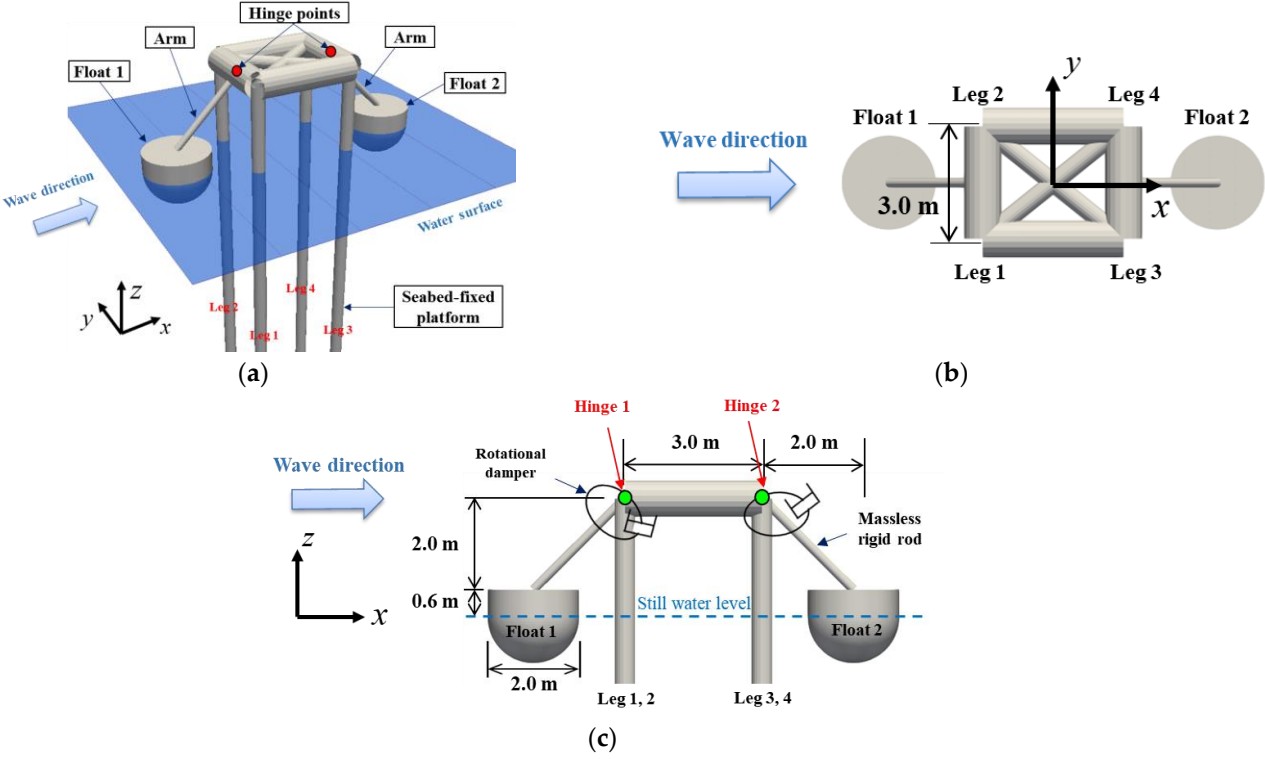

**Figure 12.** Description of the bottom-fixed platform with 2 WECs: (**a**) 3D view; (**b**) Top view; (**c**) Side view.

**Table 2.** Structural properties of the platform and float.

| Component | Description | Value | Unit |
|---|---|---|---|
| Platform | Deck size (x × y × z) | 3 × 2.4 × 2 | m |
| | Leg height | 52.6 | m |
| | Diameter of leg member | 0.45 | m |
| | Thickness of leg member | 12 | mm |
| | Inertia coefficient | 2.0 | - |
| | Drag coefficient | 1.0 | - |
| | Deck weight | 980 | kN |
| | Structural weight per unit volume | 78.6 | $kN/m^3$ |
| | Modulus of elasticity | $2.1 \times 10^8$ | $kN/m^2$ |
| | Modulus of rigidity | $8.33 \times 10^7$ | $kN/m^2$ |
| | Number of elements | 124 | - |
| | Number of nodal points | 122 | - |
| Float | Diameter | 2.0 | m |
| | Draft | 1.0 | m |
| | Mass | 2147 | kg |
| | Damping coefficient of a rotational damper | 100 | kN·m/(rad/s) |
| | Number of elements | 8520 | - |

The rotational damping proportional to the rotational velocity of the float was applied to the center of rotation as the PTO system. The optimal extraction coefficient was obtained as 100 kN·m/(rad/s) through a parametric study for the regular wave [19]. The wave power generated by the PTO system was proportional to the square of the rotational velocity of the float, as shown in the following equation.

$$P_{PTO}(t) = B_{PTO} \cdot \left\{ \dot{\theta}(t) \right\}^2 \tag{24}$$

The distance between hinge points was set to 3 m. This distance does not affect the pitch RAO of the float significantly when the platform is fixed [19]. The frequency-dependent hydrodynamic coefficients considering the interaction between the two floats were obtained using multi-body WAMIT. The interactive hydrodynamic coefficients are the same because the two floats have the same shape and size.

The natural frequency of the platform was obtained by modal analysis, and the total number of vibration modes was 708. Table 3 lists the natural frequencies up to the fifth mode. In this study, all vibration modes were considered, and the equation of motion of the platform was expressed using 708 SDOFs.

**Table 3.** Natural frequency of the platform.

|  | 1st Mode | 2nd Mode | 3rd Mode | 4th Mode | 5th Mode |
|---|---|---|---|---|---|
| Natural frequency [rad/s] | 0.4026 | 0.4059 | 3.1132 | 6.3066 | 6.3739 |
| Natural period [s] | 15.60 | 15.48 | 2.02 | 1.00 | 0.99 |

The results without WECs were first compared to identify the dynamic response characteristics of the platform. Figure 13 shows the time histories of *x*-axis displacement at the top of Leg 1 and stress at the bottom of Leg 1. The stress shown here represents the sum of axial stress and bending stress acting on each element of the structure. According to the figure, the displacement and stress increased significantly due to the influence of the floats. The response of the platform was affected by the horizontal and vertical constraint forces from the floats. The negative (compressive) stress was caused by external forces from the floats and the weight of the platform. The former particularly plays a significant role in the increase in dynamic stress. As the displacement increases, the bending stress of each element also increases, resulting in a similar trend to the time series of displacement and stress. The results underscore that this kind of fully coupled dynamic analysis is essential for a reliable estimation of the fatigue life of the system.

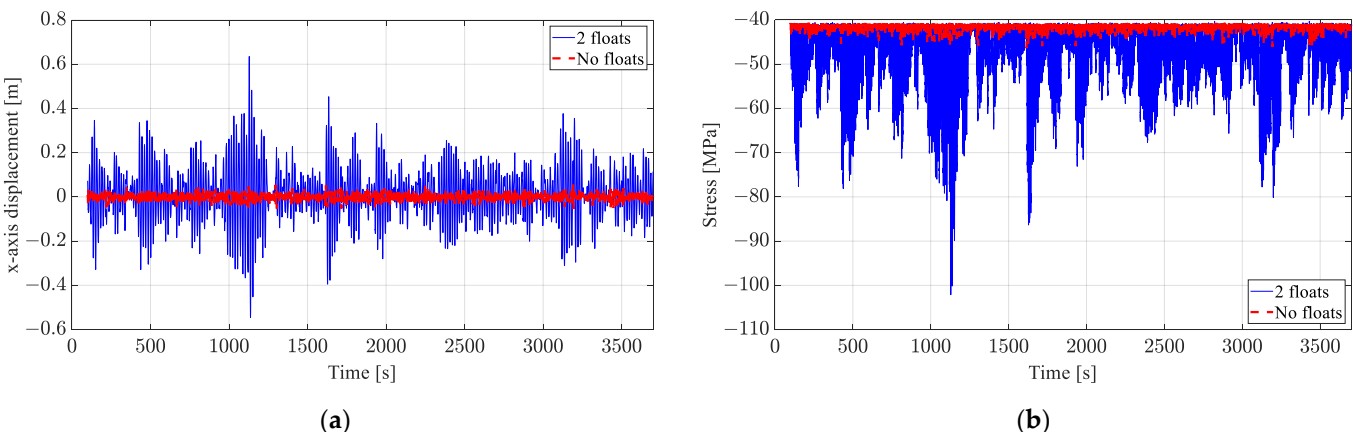

(**a**)　　　　　　　　　　　　　(**b**)

**Figure 13.** Time histories of dynamic responses of Leg 1: (**a**) *x*-axis displacement at the top; (**b**) Stress at the bottom.

Figure 14 shows the distribution of the maximum absolute value of the displacement and stress of each element from the seabed to the deck. The displacement at the top had

the same value in all legs because the stiffness of the deck was considerable. In addition, the stresses on legs 1 and 2 (front legs) are the same because the structure/loading is *x*-axis-symmetric. Similarly, the stresses on legs 3 and 4 (rear legs) were the same. The displacements were zero at the sea bottom because the structure was fixed there and increased monotonically toward the top, as shown in Figure 14a. The top displacement was increased significantly (~10 times) with the two floats compared to the no-float case. The maximum stresses occurred at the top and the bottom of the foundation structure. The significantly increased stress at the top was attributed to the dynamic coupling with the WECs, as shown in Figure 14b. The magnitude of the stress was the smallest in the middle of the leg because of the smallest degree of bending. The rear legs had a maximum stress approximately 7% greater than the front legs because of the differences in interaction forces by front and rear floats.

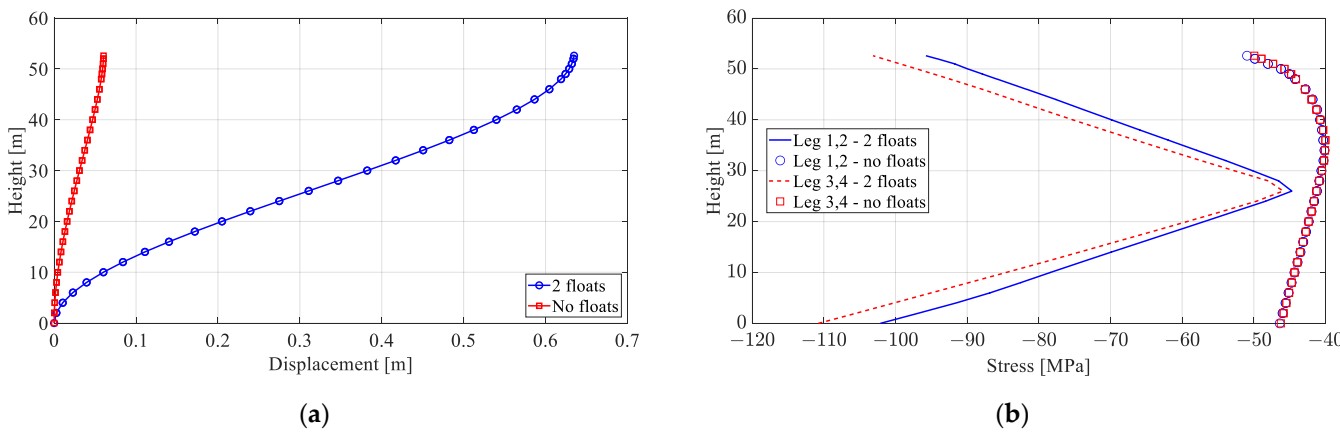

(**a**)    (**b**)

**Figure 14.** Maximum responses along the height of the platform: (**a**) Displacements; (**b**) Stresses.

Table 4 lists the statistics of the dynamic responses and stresses of the platform. Owing to the influence of WECs, the positive maximum x-displacement at the top of the structure was larger than the negative maximum. This means that the structure tends to move more in the incident wave direction. In this study, simulation results were analyzed using Root Mean Square (RMS), one of the statistical parameters as a measure of dynamic fluctuation. The RMS values of the stress were approximately 20% increased because of the influence of the floats. On the other hand, the maximum value differed by approximately 2.2 times. Hence, fully coupled analysis is essential for the reliable design of the structure.

**Table 4.** Statistics of the dynamic responses of the platform.

| Parameter | Case | Maximum | Minimum | RMS | Standard Deviation |
|---|---|---|---|---|---|
| *x*-axis displacement at the top [m] | 2 floats | 0.635 | −0.546 | 0.122 | 0.121 |
|  | No floats | 0.059 | −0.060 | 0.014 | 0.014 |
| Stress at the bottom [MPa] | 2 floats—Leg 1,2 | −40.31 | −102.11 | 50.99 | 8.02 |
|  | 2 floats—Leg 3,4 | −39.54 | −110.74 | 51.17 | 8.23 |
|  | No floats—Leg 1,2 | −40.61 | −46.39 | 41.85 | 0.78 |
|  | No floats—Leg 3,4 | −40.53 | −46.23 | 41.82 | 0.73 |

Figure 15 and Table 5 show the time histories of the constraint forces acting on the hinge points and their statistics considering the influence of the floats. As shown in Figure 15a, the *x*-axis constraint force at Hinge 1 (front) has large positive values, and that at Hinge 2 (rear) has similar negative values. This is because the movement of each float is constrained by the hinge point, so the constraint force acts in the opposite direction to the movement of the float to satisfy the constraint condition. The *z*-axis constraint forces acting on the two hinge points are similar in magnitude, as shown in Figure 15b. The RMS value of the constraint force is larger in the *z*-axis, but the maximum value occurred on the *x*-axis.

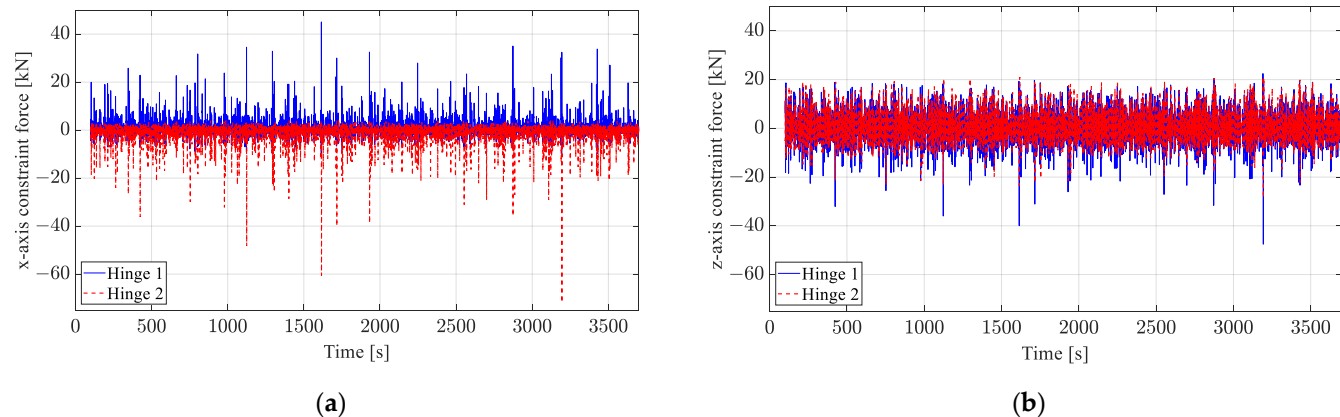

**Figure 15.** Time histories of the constraint forces acting on the hinge points: (**a**) *x*-axis; (**b**) *z*-axis.

**Table 5.** Statistics of the constraint forces acting on the hinge points.

| Parameter | Float | Maximum | Minimum | RMS | Standard Deviation |
|-----------|-------|---------|---------|-----|--------------------|
| *x*-axis constraint force [kN] | Float 1 | 45.05 | −9.97 | 2.90 | 2.78 |
| | Float 2 | 4.33 | −71.79 | 3.49 | 3.29 |
| *z*-axis constraint force [kN] | Float 1 | 22.48 | −47.53 | 6.70 | 6.66 |
| | Float 2 | 21.18 | −28.50 | 6.14 | 6.07 |

Figure 16 and Table 6 show the differences in the float rotational motions with or without considering platform elastic responses, i.e., elastic platform vs. fixed platform. The rotational motions of both floats are slightly increased after including the elastic response of the platform. Because the foundation structure is not as flexible, their differences are expected to be small, as in Figure 16 and Table 6.

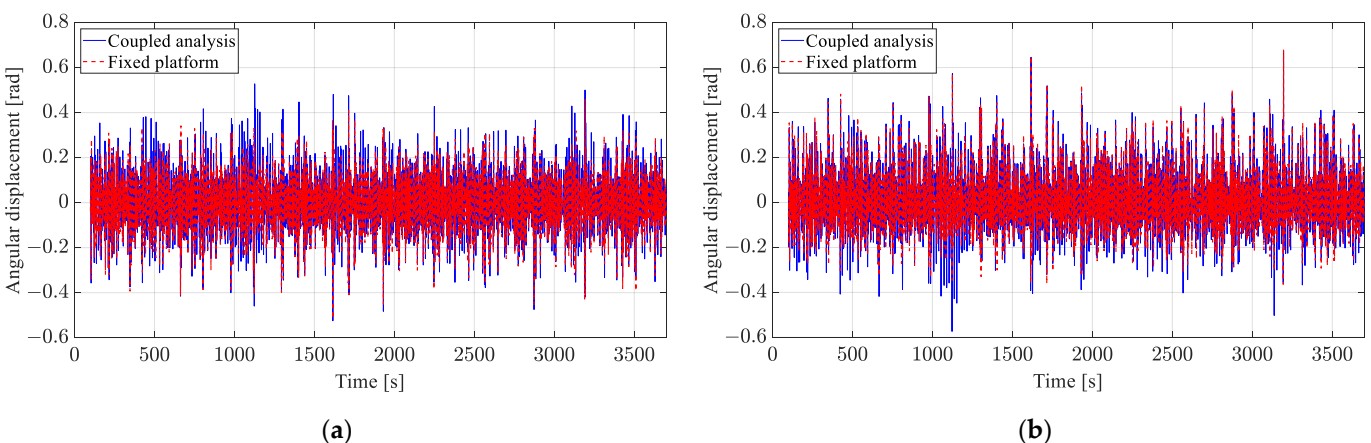

**Figure 16.** Time histories of rotational displacements of the floats: (**a**) Float 1; (**b**) Float 2.

**Table 6.** Statistics of the rotational displacement of the floats (unit: rad).

| Case | Float | Maximum | Minimum | RMS | Standard Deviation |
|------|-------|---------|---------|-----|--------------------|
| Coupled analysis | Float 1 | 0.528 | −0.525 | 0.126 | 0.126 |
| | Float 2 | 0.673 | −0.574 | 0.134 | 0.134 |
| Fixed platform analysis | Float 1 | 0.460 | −0.515 | 0.106 | 0.106 |
| | Float 2 | 0.679 | −0.365 | 0.118 | 0.118 |

Figure 17 and Table 7 show the time histories of the produced wave power and their statistics for both fixed and elastic foundation structures. Similar to the previous results, the RMS values of the produced power from Floats 1 and 2 are affected slightly by the

elastic responses of the platform. Interestingly, the average power production of Float 2 (rear) was slightly larger than that of Float 1 (front) because of two-body hydrodynamic interactions (see Figure 9) and other coupled effects. The power fluctuations (standard deviation) were reduced slightly, which is beneficial, by considering coupled effects with the platform elasticity.

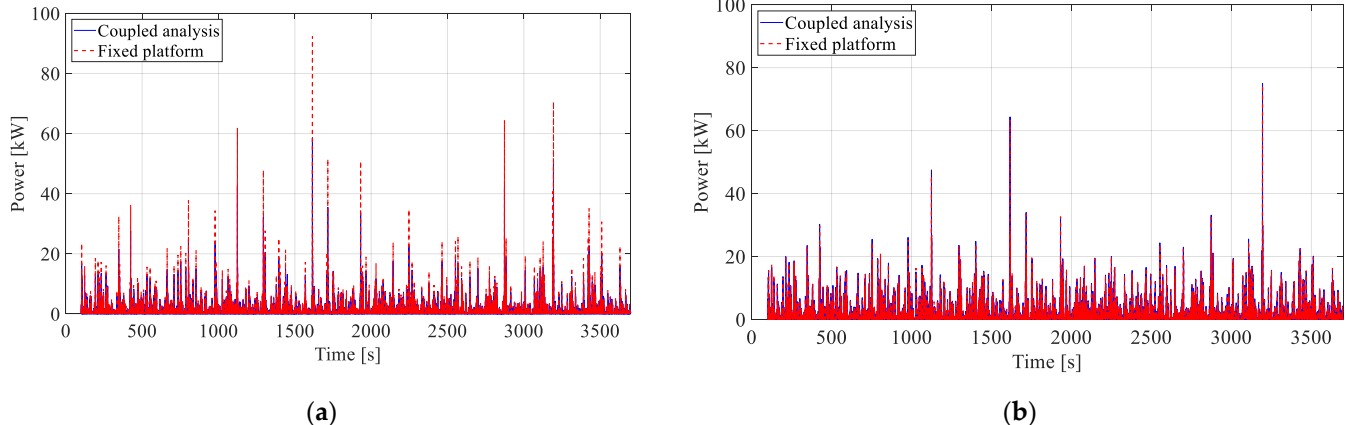

**Figure 17.** Time histories of produced wave power: (**a**) Float 1; (**b**) Float 2.

**Table 7.** Statistics of the produced wave power (unit: kW).

| Case | Float | Average | Maximum | RMS | Standard Deviation |
|---|---|---|---|---|---|
| Coupled analysis | Float 1 | 1.847 | 58.62 | 3.34 | 2.78 |
| | Float 2 | 2.206 | 74.98 | 4.14 | 3.50 |
| Fixed platform analysis | Float 1 | 1.862 | 92.32 | 3.81 | 3.32 |
| | Float 2 | 2.169 | 74.53 | 4.10 | 3.48 |

## 4. Conclusions

This study developed a numerical procedure for the coupled dynamic response analysis between two floats, connecting arms with hinge points, and a bottom-fixed elastic foundation structure under irregular wave conditions. The foundation structure was modeled using three-dimensional beam elements, and its natural frequencies and vibration modes were obtained through modal analysis. The equation of motion of the structure was decomposed into multiple SDOFs using a modal matrix to improve the calculation time. The motion of the floating body was analyzed using an augmented formulation, one of the multi-body dynamics techniques. The frequency-dependent hydrodynamic coefficients of the dual floating body were obtained using WAMIT, a hydrodynamic solver. The float was modeled with many quadrilateral elements, and the body-nonlinear Froude–Krylov based on linear wave theory and hydrostatic forces were calculated with respect to the instantaneous wetted body surface at every time step. The respective steps of the present hydrodynamics results were compared with the published results, and they showed good agreement.

Using the developed time-domain program, the fully coupled dynamics of a dual-buoy Wavestar WECs connected to an elastic foundation structure were investigated for a typical irregular wave condition in a Korean nearshore region. First, after coupled dynamic analysis considering the influences of the WECs, the maximum values of the displacement and stress of the foundation structure were increased 10- and 2.2-fold, respectively. The RMS values of the stresses were also increased by approximately 20% by the influence of the WECs. The results underscore that this kind of fully coupled dynamic analysis is essential for a reliable estimation of the fatigue life of the system. The increased dynamic responses of the foundation structure directly affect the forces transmitted to the WECs, but they have little influence on the motion of the floats and the corresponding generated power outputs.

Interestingly, the average power production of Float 2 (rear) was slightly larger than that of Float 1 (front) because of two-body hydrodynamic interactions and other coupled effects. Considering the coupled effects with platform elasticity, the power fluctuation (standard deviation) is reduced slightly, which benefits the electric system. Overall, the coupled dynamic analysis is essential to the reliable performance evaluation and design of such a system.

**Author Contributions:** Conceptualization, S.H. and W.K.; methodology, S.H. and W.K.; software, S.H. and W.K.; validation, S.H.; formal analysis, S.H. and W.K.; investigation, S.H. and W.K.; writing—original draft preparation, S.H.; writing—review and editing, S.H., M.-H.K. and W.K.; supervision, M.-H.K. and W.K.; project administration, W.K.; funding acquisition, W.K. All authors have read and agreed to the published version of the manuscript.

**Funding:** This work was supported by INHA UNIVERSITY Research Grant.

**Institutional Review Board Statement:** Not applicable.

**Informed Consent Statement:** Not applicable.

**Conflicts of Interest:** The authors declare no conflict of interest.

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
