# Peer review of "Coupled Dynamic Analysis of a Bottom-Fixed Elastic Platform with Wave Energy Converters in Random Waves"

_applsci, doi:10.3390/app12157915_

Round 1

Reviewer 1 Report

The article is an interesting example of combining the modeling process with the optimization of real object parameters. The authors carefully discussed the analyzed issue, presented the model and the results of simulation and optimization in relation to the experimental data. The work is written coherently, logically and consistently. In the opinion of the reviewer, the following issues require clarification and supplementation:

1. The assumptions made in the construction of the mathematical model should be collected and clearly presented, for example in points.
2. Equation (18) requires explanation or a reference from the literature.
3. The acronym RAO should be explained at the beginning of the text for readers who are not specialists in this field.
4. The parameters and methods of the numerical process should be specified in detail.
5. The authors provide RMS values ​​for stress, linear displacement, force, angular displacement and power. Do these values ​​have a physical sense? Or is it just a statistical parameter?

After these explanations and supplements, the article may be published, in the opinion of the reviewer.

Author Response

Please see the attached file for author's response

Reviewer 2 Report

This paper is about a point-absorber-type wave energy converter on an elastic foundation structure. The performance of the system under a selected sea condition is evaluated. The topic of the paper falls within the scope of this journal. My specific comments are as follows: 

My specific comments are as follows: 

1.     The novelty of this work is not clear. The dynamic response and power production of the Wavestar WEC have been conducted in various studies. Such topic is not new.

2.     The derivation of Eqs. (2), (3) and (4) should be given. If they are not derived by the authors, it is better to add some references and discuss the conditions that they can be used.

3.     What is the physical meaning of Qd?

4.     The definition of the constraint force vector acting on the joint is defined in Eq. (3). However, the constraint force has not been involved in the following studies. It suggests that it is not a fully coupled dynamics analysis.

5.     “... The linear hydrostatic force is proportional to the vertical displacement from the still water level regardless of changes in water level ...”

This is true for structures which are wall-sided near the free surface. The authors discuss that such statement is still true for structures whose submerged part is a hemisphere.

6.     “... Nonlinear Froude-Krylov forces must be considered to perform a precise dynamic analysis of HPA-type WECs ...”

It is normally understood that the wave diffraction plays an important role in the analysis of wave action with large-scale structures. However, the nonlinear wave diffraction effect has been ignored by the authors, while only the nonlinear Froude-Krylov forces are included.

7.     Eq. (1) gives the equation of body motion in the time domain. However, Eq. (1) is developed for a single body, while not applicable to a multi-body system. The coupling effects between the motions of different bodies have not been taken into account, which may not provide reliable results.

8.     The authors stated the nonlinear Froude-Krylov forces are used; however Eq. (12) gives the linear incident potential;

9.     There is a lack of the derivation of Eq. (10).

10.   In the abstract: “...The program included the FE (Finite Element) model of elastic foundation, the multi-body dynamics model of the entire system, the hydrodynamics model of the dual-buoy, and fully coupled dynamics ...”

The finite element model have not been considered in this study. The coupling between different components of the system has not been discussed. The abstract does not reflect the works that the author have done, misleading the readers.

11.   The validation is carried out for different cases of single bodies. The validation should also be conducted for dual-body systems.

Author Response

Please see the attached file for authors' response

Round 2

Reviewer 2 Report

The paper in the present form may reach the level of JMSE.